# Can LLMs predict the convergence of Stochastic Gradient Descent?

Oussama Zekri [1 2]   Abdelhakim Benechehab [1]   Ievgen Redko [1]

## Abstract

Large-language models are notoriously famous for their impressive performance across a wide range of tasks. One surprising example of such impressive performance is a recently identified capacity of LLMs to understand the governing principles of dynamical systems satisfying the Markovian property. In this paper, we seek to explore this direction further by studying the dynamics of stochastic gradient descent in convex and non-convex optimization. By leveraging the theoretical link between the SGD and Markov chains, we show a remarkable zero-shot performance of LLMs in predicting the local minima to which SGD converges for previously unseen starting points. On a more general level, we inquire about the possibility of using LLMs to perform zero-shot randomized trials for larger deep learning models used in practice.

Figure 1. Overview of the proposed approach. After having run SGD on a given optimization problem, we tokenize the obtained iterates and feed them to an LLM of choice. We further use the logits to fill the transition kernel of the Markov chain underlying the SGD with probabilities $P(x_i|x_j)$, while imputing those of its elements that were not observed. Finally, we use the estimate transition kernel to do forecasting for previously unseen inputs.

## 1. Introduction

The research in machine learning (ML) and artificial intelligence (AI) fields has recently exhibited a drastic advancement with several breakthrough results across a wide range of tasks. The paramount of it is undoubtedly represented by the introduction of large language models (LLMs) (Brown et al., 2020; Touvron et al., 2023): the most powerful AI models currently available. Trained on the vast amounts of language data, LLMs have achieved state-of-the-art results in diverse applications including machine translation (Brown et al., 2020), text generation, question answering (Roberts et al., 2020), and sentiment analysis (Zhang et al., 2023a). One of the reasons that makes studying LLMs so fascinating is the remarkable zero-shot performance often seen as a sign of their emergent capabilities.

One particular example of LLMs zero-shot capabilities that recently gained in popularity is their highly competitive

performance in time-series forecasting (Jin et al., 2024). The cornerstone idea enabling their use in this task is to represent time series data in a textual format through careful tokenization (Gruver et al., 2023). Several works have used it as a foundation to rival dedicated time-series forecasting models with very encouraging results. More interestingly, a recent paper (Liu et al., 2024) applied such tokenization to tackle a completely different task consisting in (in-context) learning of the transition probabilities of the dynamical systems that time series data describe. The intuition behind such an approach was to treat logits of the LLM's next-token prediction output as the above-mentioned transition probabilities and refine them to a desired degree of accuracy depending on the chosen discretization.

While presenting intriguing results related to different dynamical systems, their work doesn't provide an actionable way to use the derived transition probabilities. Similarly, their work concentrates on well-known illustrative examples of dynamical systems, that – while being insightful – do not correspond to ML tasks solved in practice.

**Contributions** In this paper, we propose to substantially expand the scope of (Liu et al., 2024). Our contributions in this direction are as follows:

---

[1]Huawei Noah's Ark Lab [2]ENS Paris-Saclay. Correspondence to: Oussama Zekri <oussama.zekri@ens-paris-saclay.fr>.

*Proceedings of the 1st Workshop on In-Context Learning at the 41st International Conference on Machine Learning*, Vienna, Austria. 2024. Copyright 2024 by the author(s).

1. We consider a challenging task of understanding the dynamics of the stochastic gradient descent in convex and non-convex settings with LLMs;

2. By leveraging the theoretical link between Markov chains and SGD, we propose an algorithmic way not only to retrieve the transition probabilities of the Markov chain underlying the SGD, but also to estimate its transition kernel.

3. We provide preliminary experimental results showing the efficiency of the transition kernel estimation and its application in predicting the convergence of SGD from previously unseen random initialization.

The rest of our paper is organized as follows. In Section 2, we present the details about the prior work on the (in-context) learning of the transition probabilities of dynamical systems. Section 3 presents the details regarding the equivalence of the stochastic gradient descent to Markov chains and our approach to estimating the transition kernel of the latter. In Section 3.3, we present the experimental results showcasing the ability of our approach to correctly estimate the transition kernel of toy Markov chains and its application to SGD for both convex and non-convex optimization problems. Finally, we conclude in Section 4.

## 2. Background knowledge

**In-context Learning (ICL)** is a growing research field aiming at improving the zero-shot capabilities of LLMs by using a carefully designed context included in the prompt. Since its introduction by (Brown et al., 2020), ICL has been successfully used in many practical applications including NLP (Wei et al., 2022; Yao et al., 2023), vision (Dong et al., 2023; Zhang et al., 2023b; Zhou et al., 2024), and time series forecasting (Gruver et al., 2023; Jin et al., 2024).

**ICL with dynamical systems** In our work, we are particularly interested in a recent study (Liu et al., 2024) investigating the inference of transition probabilities of known dynamical systems from simulated trajectories using ICL. The authors of the above-mentioned work, show that medium-size LLMs, such as LLaMA2-13B, are able to learn the dynamics of *Markovian* systems with various properties (e.g. chaotic, discrete, continuous, stochastic).

More formally, for a time series $(x_i)_{i \leq t}$ generated by simulating a dynamical system with predefined transition rules[1], Liu et al. (2024) apply the following procedure to infer them conditioned on the observed states $x_i$ (see Appendix A for more details):

---

[1]E.g. Brownian motion: $X_{t+1}|X_t = x_t \sim \mathcal{N}(x_t + \mu, \sigma^2)$ with parameters $(\mu, \sigma)$ or discrete Markov chains with $n$ states: $P_{ij} = P(X_{t+1} = j|X_t = i)$ for $1 \leq i, j \leq n$: $P(X_{t+1}|X_t)$

---

**Procedure:** ICL for dynamics learning

**Input:** time serie $(x_i)_{i \leq t}$, LLM $M$, precision $k$
1. Rescale and encode the time serie with $k$ digits

$$\hat{x}_t = "x_1^1 x_1^2 ... x_1^k, ..."$$

2. Call $M(\hat{x}_t)$
3. Extract the digits logits $(0, 1, 2, 3, \ldots, 9)$
4. Build the next state probability distribution using the Hierarchy-PDF algorithm in (Liu et al., 2024)
**Return:** predicted transition rules for the observed states: $\{P(X_{i+1}|X_i = x_i)\}_{i \leq t}$

---

We now present our main contributions.

## 3. LLMs understand the convergence of SGD

### 3.1. Problem setup

Given a training set $x = (x_1, \ldots, x_N)$ of $N$ i.i.d samples, we consider the optimization of the following problem

$$\min_\theta F(\theta), \quad F(\theta) = \frac{1}{N} \sum_{i=1}^N f(x_i, \theta), \quad (1)$$

where $\theta \in \mathbb{R}^d$. For this problem, the updates of minibatch SGD with stepsize $\gamma_t$ and mini-batch $B_t$ of size $m$ have the following form

$$\theta^{t+1} = \theta^t - \gamma_t \nabla \tilde{f}_t(\theta^t) \quad (2)$$

where $\theta^t$ denotes the parameters after $t$ iterations, and $\nabla \tilde{f}_t(\theta^t) = \frac{1}{m} \sum_{x \in B_k} \nabla_\theta f(x, \theta^t)$ where $B_t$ is a minibatch of size $m$ of training examples selected randomly.

### 3.2. Overparametrized vs. underparametrized regime

Our main underlying idea is to rely on the equivalence between the SGD and the Markov chain established in (Bach & Moulines, 2013) and used in several other works to theoretically analyze SGD. More formally, (Dieuleveut et al., 2018) took advantage that for a fixed constant step-size $\gamma_t = \gamma$, the SGD updates (2) form a homogeneous Markov chain. This Markov chain converges to a unique stationary distribution $\pi_\gamma$ that depends on the regime of the ML problem. In the *overparametrized* regime, i.e. when $d$ is larger than $N$, the Markov chain converges to a Dirac $\pi_\gamma = \delta_{\tilde{\theta}^*}$ where $\tilde{\theta}^*$ is a specific solution that depends on many parameters (e.g. initialization, step-size, model's architecture etc.). In the *underparametrized* regime, $\pi_\gamma$ is a stationary distribution with a strictly positive variance, e.g. $\mathcal{N}(\theta^*, \gamma^{1/2})$ where $\theta^*$ is an optimum (we note, however, that in general, the SGD noise is not Gaussian (Panigrahi et al., 2019)).

We now illustrate that a LLaMA2-7B model can understand the convergence of the SGD and correctly identify the

regime in which the iterates were obtained. For this, we consider toy underparametrized and overparametrized linear regression optimization problems in $\mathbb{R}^2$ and plot the logit probabilities outputted by the LLM given a time series of 1000 iterates in Figure 2. The time series length is selected to ensure that its tokenized representation remains within the LLM's context window limit, and we set the temperature $T$ of the LLM to 1. We can see that in both cases the LLM correctly identifies the regime by either outputting logits that form a Dirac distribution for the overparametrized problem or a Gaussian-like distribution with an accurately estimated mean and covariance of the underparametrized case.

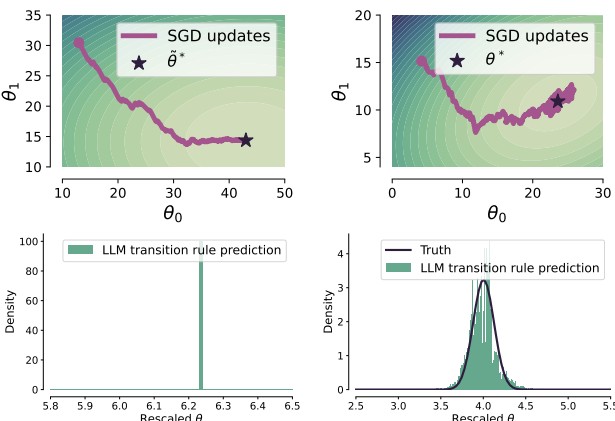

*Figure 2.* **Top Left** and **Top Right**, a run of SGD in the overparameterized and underparameterized regimes, respectively. **Bottom Left** and **Bottom Right**, transition probabilities predicted by LLM in overparameterized and underparameterized regimes.

### 3.3. From understanding to forecasting

Similarly to (Liu et al., 2024), we now know that LLMs can understand the SGD in two different regimes. We now want to make a step further by finding a way to benefit from this knowledge. One tangible way for this is to use the transition probabilities to estimate the transition kernel of the Markov chain underlying the SGD. Once this is achieved, it can be used to do forecasting simply by running the Markov chain on new previously unseen inputs representing, for instance, different initilization points or seeds.

**Estimating the transition kernel of SGD** Since SGD is an infinite-dimensional state-space Markov chain, we propose a method to estimate a discretization of its transition kernel. For each parameter $\theta_i, i \in \{1, \ldots, d\}$, we consider the discretized state vector $\Theta_i^t$ at time $t$, i.e. the vector $\Theta_i^t = (0, \ldots, 0, 1, 0, \ldots, 0)^\top$ with a 1 at the $l$-th position if $\Theta_i^t$ is in state $l$ at time $t$. This is a vector of size $10^k$, where $k$ is the chosen precision. Then, we can write $\Theta_i^{t+1} = \sum_{j=1}^{d} \lambda_{i,j} P^{(i,j)} \Theta_j^t$, where $\forall i, j, \lambda_{i,j} \geq 0, \sum_{j=1}^{d} \lambda_{i,j} = 1$ and $P^{(i,j)}$ is the discretized transition probability matrix for the transitions of states of parameter $\theta_j$ to states of

parameter $\theta_i$, (Ching et al., 2002). Then, the discretized transition kernel of SGD can be seen as a matrix

$$
Q = \begin{pmatrix} \lambda_{1,1} P^{(1,1)} & \cdots & \lambda_{1,d} P^{(1,d)} \\ \vdots & \ddots & \vdots \\ \lambda_{d,1} P^{(d,1)} & \cdots & \lambda_{d,d} P^{(d,d)} \end{pmatrix}
$$

which satisfies $\Theta^{t+1} = Q\Theta^t$. Our method estimates the matrix $P^{(i,i)}$, from a single observation of the time serie of the $i$-th parameter $\theta_i$.

The LLM predictions help us to fill a few rows of $P^{(i,i)}$. To completely fill this sparse matrix, we compute the debiased Sinkhorn barycenter of the distributions surrounding the empty rows (Janati et al., 2020; Flamary et al., 2021), see Figure 1 for the overview of the whole pipeline and Algorithm 1 for the transition kernel estimation routine. In

---

**Algorithm 1** Estimating $P^{(i,i)}$

**Input:** time serie $(\theta_i^{t+1})_{t \geq 0}$, LLM $M$, precision $k$, regularization $\varepsilon$
1. Fill $s < 10^k$ rows of the $10^k$ rows of $P^{(i,i)}$ with Procedure$(\theta_i^{t+1}, M, k)$, denoted as $(P_1^{(i,i)}, \ldots, P_s^{(i,i)})$
2. Fill the remaining $10^k - s$ rows of $P^{(i,i)}$ with debiased Sinkhorn barycenter of regularization parameter $\varepsilon$ :
**for** $j = 1$ **to** $s - 1$ **do**
  **if** empty rows between $P_j^{(i,i)}$ and $P_{j+1}^{(i,i)}$ **then**
    Compute debiased Sinkhorn barycenter between $P_j^{(i,i)}$ and $P_{j+1}^{(i,i)}$, with regularization parameter $\varepsilon$
    Fill the empty rows
  **end if**
**end for**
**Return:** Estimated matrix $P^{(i,i)}$

---

practice, estimating the correlation matrices $P^{(i,j)}$ for $i \neq j$ is hard as it requires considering a multivariate Markov chain (Ching et al., 2002). We leave this generalization for future work, although our experimental result suggest that estimating only the block matrices in the diagonal of $Q$ (i.e., assuming $\lambda_{i,j} = 0$ for $i \neq j$) may be enough to obtain a reasonable estimate of $Q$.

**Predicting SGD convergence with LLMs** We now consider a convex and a non-convex optimization problem to illustrate the usefulness of our approach.

#### 3.3.1. CONVEX CASE

We consider a usual linear regression problem in $\mathbb{R}^2$. We start by performing one SGD run with constant step-size $\gamma$ and use Algorithm 1 to estimate the transition matrices $P^{(1,1)}$ and $P^{(2,2)}$ of parameters $\theta_1$ and $\theta_2$.

Using the estimated matrices, we then show in Figure 3 that running a Markov chain with $Q$ on new starting points leads

to the convergence to the global optimum. The latter behavior reflects our accurate estimation of the transition kernel that replaces gradient computations with computationally cheap matrix multiplications.

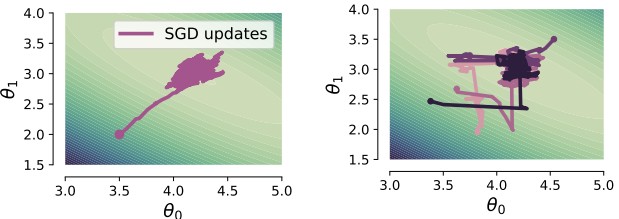

*Figure 3.* We optimize $F$ defined in (1) with $f(x_i, \theta) = \frac{1}{2}(\langle x_i, \theta \rangle_{\mathbb{R}^2} - y)^2$ for $d = 2$ and $N = 100$ (see more instances in Appendix D). **Left.** A full SGD run in the convex case. The visited states constitute the time serie shown to the LLM to estimate the transition kernel. **Right.** Starting from different initial points, simulating the convergence of the SGD with the estimated transition matrix leads to convergence to the same global minima.

### 3.3.2. NON-CONVEX CASE

For the non-convex case, we do the same experiment, but this time we launch two SGD runs with the same constant step-size $\gamma$ and different initial points. The two runs are not trapped in the same optimum valley allowing to better estimate the transition kernel, see Figure 4.

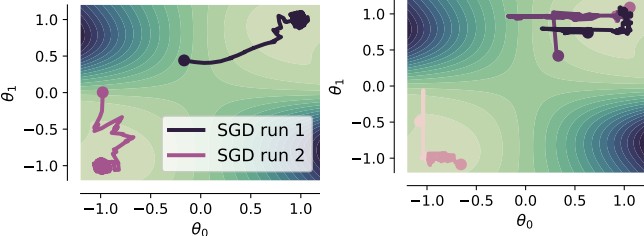

*Figure 4.* We optimize $F$ defined in (1) with $f(x_i, \theta) = \frac{1}{2}(\theta_0 \sin(\theta_1 x_i) - y)^2$ for $d = 2$ and $N = 100$. **Left.** A full SGD run in the non-convex case. The visited states constitute the time serie shown to the LLM and used to estimate the transition kernel. **Right.** Starting from different initial points, we run the Markov chain with the estimated transition kernel and converge to the same local minima as SGD.

### 3.4. ICL neural scaling laws revisited

We end this short paper by providing an important insight into the neural scaling laws of ICL derived by the authors of (Liu et al., 2024). In their paper, the authors argue that ICL exhibits power scaling laws similar to those of training (Kaplan et al., 2020). Additionally, they add that for some dynamical systems, one observes plateauing effect suggesting that it happens when the dynamical system "wander out" and doesn't converge to a stationary distribution.

We provide a different point of view on their analysis using our proposed framework. For simplicity, we consider a Markov chain with 2 states, for which the transition matrix $P = (P_{ij})_{1 \leq i,j \leq 2}$ is defined as

$$ P = \begin{pmatrix} p & 1-p \\ 1-q & q \end{pmatrix} $$

with $p, q \in (0, 1)$.

The spectrum of this homogeneous, reversible, aperiodic, and irreducible Markov chain is $\text{Sp}(P) = \{1, p+q-1\}$. The spectral gap $\rho = 1 - |p + q - 1|$ gives us information on its speed of convergence. For this type of Markov chain, for any initial law $\pi_0$, we have that $d_{TV}(\pi_t, \pi) \leq C_\pi \exp(-\rho t)$, where $d_{TV}$ is the total variation distance, $\pi_t$ is the distribution at time $t$, $\pi$ is the stationary distribution of the Markov Chain and $C_\pi$ is some constant term that depends on $\pi$. See (Pitman & Hough, 2003)[Theorem 28.5].

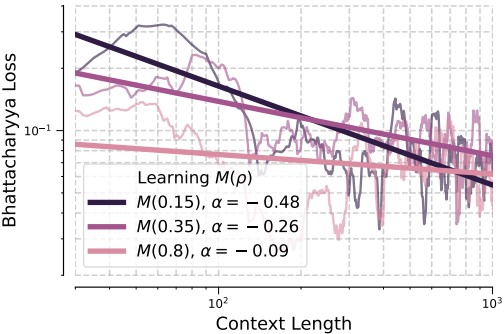

*Figure 5.* Neural scaling laws for different values of $\rho$. $M(\rho)$ denotes a 2-state Markov chain of spectral gap $\rho$.

In Figure 5, we observe that the spectral gap influences the coefficient of the power law underlying the neural scaling law of ICL. This is contrary to what is claimed in (Liu et al., 2024) as the studied Markov chain admits a stationary distributions for all studied values of $p$ and $q$. In particular, it suggests that it is easier for the LLM to understand the transition probabilities of a Markov chain that tends more slowly toward its invariant distribution.

## 4. Conclusion

In this work, we extend the ICL abilities of LLMs to a realistic and challenging problem: estimating the transition kernel of SGD. We show the feasibility of this task by providing a systematic way to generalize the learned kernel to previously unseen states, both in convex and non-convex optimization landscapes. The most important open question that stems from this work is whether such an approach can be applied to ML models with orders of magnitudes more parameters and how to raise the computational challenge underlying this potentially highly impactful task.

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

# Appendix

## Table of Contents

## A. Detailed ICL for dynamics learning

In this section, we go though the *procedure* presented in Section 2, providing more details about each step of the process. Given a time series $\{x_1, x_2, \ldots, x_t\}_{t>>1}$, the ICL for dynamics learning procedure presented in (Liu et al., 2024) goes as follows:

1. **Rescaling.** To avoid amibiguity due to leading zeros or leading same digit, the values of the time serie elements are rescaled to the interval $[1.5, 8.5]$. E.g. $[0.2513, 5.2387, 9.7889] \rightarrow [1.5, 5.16, 8.5]$

2. **Fixed-precision encoding.** Represent the time series elements with a fixed precision of $k$ digits. E.g. $[1.5, 5.16, 8.5] \rightarrow [150, 516, 850]$ with $k = 3$

3. **String representation.** Represent the time serie as a string. E.g. $[150, 516, 850] \rightarrow$ "150, 516, 850"

4. **Tokenization.** Transform the string using the LLM's corresponding tokenizer (see Appendix B for more details). E.g. "150, 516, 850" $\rightarrow [29896, 29945, 29900, 29892, ...]$

5. **Inference.** Call the LLM to produce logits over the full tokens vocabulary. $LLM([29896, 29945, 29900, 29892, ...] \in \mathbb{R}^L) \rightarrow logits \in \mathbb{R}^{L \times N_t}$ with $N_t$ the vocabulary size and $L$ the sequence length.

6. **Softmax.** Extract the logits corresponding to single digits $(0, 1, 2, \ldots, 9)$ and apply the Softmax to get a probability distribution over the latters. E.g. $logits \in \mathbb{R}^{L \times N_t} \rightarrow probs \in \mathbb{R}^{L \times 10}$

7. **Hierarchy-PDF.** The trivial way to proceed is to sample the next digit from the obtained $probs$, and repeat step 5 in an autoregressive fashion. However, Liu et al. (2024) provide a more sophisticated algorithm that explores the modes (and their neighborhoods) of the generated $probs$. This algorithm -*Hierarchy-PDF*- allows us to build a more refined probability distribution over the desired next value with $k$ digits. E.g. $Hierarchy-PDF(serie, LLM, precision) \rightarrow probs \in \mathbb{R}^{L \times 10^k}$

8. **Transition rule.** The last element of the obtained $probs$ constitutes the transition rule $P(X_{t+1} = i | X_t = x_t)$ for $i \in [0, 1, 2, ..., 10^k - 1]$ in the finite-discrete space formed by steps 1 and 2.

## B. On the importance of the tokenizer

The time serie tokenization step is a crucial part of the above procedure. Indeed, LLMs' ability to handle numerical values has been proved to be dependent on the tokenization algorithm (Singh & Strouse, 2024; Ali et al., 2024; Gruver et al., 2023). The most widely used tokenization algorithm to-date, *BPE* (Sennrich et al., 2016), tend to assign tokens to arbitrary 3-digits numbers based on their occurences in large-scale corpora, and the tokenizer's vocabulary size. As highlighted by Gruver et al. (2023), this artifact severly hinders LLMs' ability to predict numerical values in-context. This is the case for popular LLMs such as *GPT-3* (Brown et al., 2020) or *Claude v2.1*.

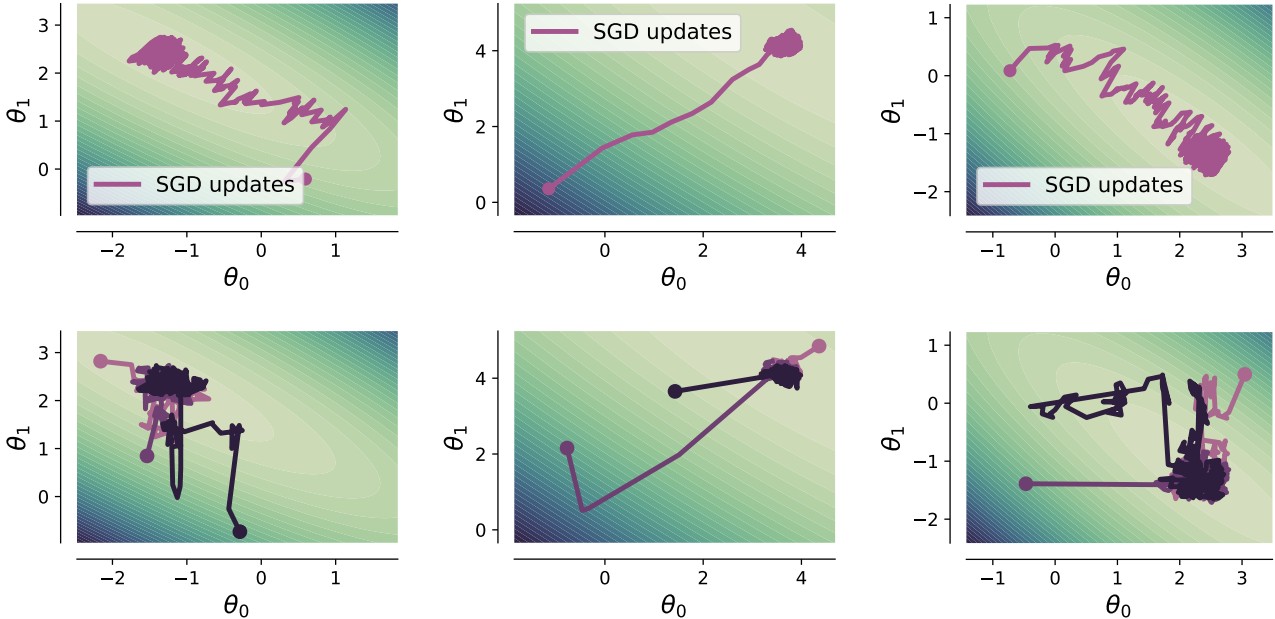

*Figure 6.* **Top**, 3 randomly generated convex problems, with one gradient descent per problem, used to learn the transition matrix $Q$ for each problem. **Bottom**, the same 3 problems, with Markov chains generated according to the learned $Q$ matrix for each problem, and different initialization points.

Newer models (*LLaMA-3*, *GPT-3.5*, *GPT-4*) however, tend to have hard-coded rules on top of *BPE*, making them able to encode all 3-digits numbers with their own token. Although this feature would accelerate the ICL procedure by eliminating the need for the *Hierarchy-PDF* algorithm, our first experiments show that it fails due to the under-representability of larger numbers in the training data. Indeed, we found that tokens corresponding to numbers beyond 10, are almost always assigned a near 0 probability.

For all these considerations, we decided to stick to models that only encode single digits as separate tokens, a useful feature for our goal of estimating Markov chains transition kernels. In practice, we conduct our experiments using the *LLaMA-2 (7B)* model (Touvron et al., 2023). The choice of this relatively *small LLM* further highlights the potential of our method, that can scale with newer and more expressive models (*LLaMA-3 (8B, 70B, 400+B)*). Furthermore, other tokenization techniques that are numerical values-focused has been presented in the literature (Golkar et al., 2023; Wu et al., 2024), paving the way for another research direction that may benefit our method.

## C. Obtaining ground truth for SGD

An other way to write the scheme (2) is to define the *zero-mean noise* $\xi_k$ as :

$$\xi_k(\theta) = \nabla F(\theta) - \nabla \tilde{f}_k(\theta)$$

So that we can rewrite the scheme as:

$$\theta^{k+1} = \theta^k - \gamma_k \nabla F(\theta^k) + \gamma \xi_k(\theta^k) \tag{3}$$

In (Zhu et al., 2019), with large batch size $m$, (3) is approximated by the following stochastic scheme (thanks to the Central Limit Theroem), that we call gradient Langevin dynamics (GLD).

$$\theta^{k+1} = \theta^k - \gamma_k \nabla F(\theta^k) + \gamma C_k(\theta^k) Z \tag{4}$$

where $C_k(\theta) = \sqrt{\mathbb{E}(\xi_k(\theta)\xi_k(\theta)^\top)}$ and $Z \sim \mathcal{N}(0, I_d)$.

Echoing (Panigrahi et al., 2019), the batch size $m$ needs to be large enough for the Gaussian approximation of the SGD noise to be satisfying. However, the strong point of this approximation is that it gives us ground truth. In fact, we generate the noise ourselves, so we know its mean and variance.

It should be noted that obtaining ground truths only serves to study neural scaling law, and is not at all necessary for the realization of Algorihm 1.

## D. Additional Experiments

We produced additional experiments for randomly generated underparametrized convex problems. The stepsizes $\gamma$ vary from one experiment to another, but are always constant throughout the run. See Figure 6.

