# OpenReview forum: "Can LLMs predict the convergence of Stochastic Gradient Descent?"
_ICML.cc/2024/Workshop/ICL — ICML 2024 Workshop ICL Poster_

### Official Review · Reviewer_hw9K · 2024-05-29
**Interesting methodology that warrants more contexts, theoretical justification, and experiments**

**Rating:** 2
**Fit:** 3
**Confidence:** 2

**Workshop Review:**

# Summary:
This paper builds and extends upon two prior lines of research:
1. Liu et al., 2024, which established the feasibility of using LLMs to identify Markovian process.
2. Dieuleveut et al. 2018, Bach et al., 2013, which established an equivalence between stochastic gradient descent (SGD) and markovian processes.
By combining these two perspectives, this work offers an innovative approach that uses large language models to estimate the multi-variate transition rules of stochastic gradient descent.

While I find the method of this work conceptually interesting, I think there are quite some loose ends that warrant further theoretical investigations.

# Room for improvement:

**Issue with uni-variate approximation**
Liu et al., 2024 offers a way to extract uni-variate transition functions from LLMs. However, the SGD that we care about are in very high dimensions (number of parameters). If I understand correctly, this paper assumes that each parameter of the SGD state vector can roughly be described by a uni-variate markovian process. While this is not a deal-breaker considering the method offered by this paper is approximate in spirit, I do wonder how valid this uni-variate approximation is. It is, after all, well-known that a multi-variate markovian process, when observed partially, may cease being markovian and appear to have memory.

As can be seen from Figure 3. It is insufficient to estimate only the diagonal blocks of the full transition matrix Q. While the full trajectories traverses the $\theta_0,\theta_1$-plane sideways, the simulated SGD trajectories consist mostly of vertical and horizontal trajectories, which seem like artifacts from the block-diagonal approximation of Q.

**Issue with probability distribution interpolation**
The paper proposes to interpolate the missing rows of transition matrix $P^{(i,i)}$ using "Debiased Sinkhorn barycenter" of the surrounding rows. This is assuming that the quasi-matrix representing the transition rules is, in some sense, smooth. The authors should justify this choice using some theoretical analysis or references.

**More experiments**
I like the argument on the relation between spectral gap and learning rate. However, I think the authors can make the case stronger by offering more experiments on more Markov chains with larger numbers of states. Currently, there are only three Markov chains of 2 states, and the loss curves in Figure 5 are too noisy that I wouldn't really trust the fitted scaling coefficients.

**More motivations**
I would appreciate more contexts regarding the practical value of the proposed approach. In particular, do the authors wish to accelerate or predict the training process of large models by learning its transition kernels? Not well-acquainted with this field of work, I have an intuition that this may be a quixotic endeavor: to faithfully construct the full transition matrix, one needs to thoroughly sample the state space of SGD. However, once that is done the optimization process is, in a sense, already solved.

**Reason For Not Giving Higher Score:**

Please see "room for improvement"

**Reason For Not Giving Lower Score:**

This work is relatively novel, and can be of interest to ICL, meta-learning, and optimization communities.

---

### Official Review · Reviewer_uoaC · 2024-06-03
**Cool Idea**

**Rating:** 3
**Fit:** 3
**Confidence:** 2

**Workshop Review:**

This paper proposes the cool idea of trying to see if LLMs can predict where SGD converges to. The idea is inspired by prior work that showed that models can estimate the underlying dynamics of systems when observations are provided in-context. For very simple settings, it seems that the model is able to output the transition matrix.

One thing I don't really get is how the authors are discretizing the continuous parameter space into discrete states. The number of states is huge, so it is kind of surprising to me that the LLM can actually get any kind of estimate of the transition matrix. I guess these considerations are covered in prior works but I am still curious.

**Reason For Not Giving Higher Score:**

N/A

**Reason For Not Giving Lower Score:**

The idea is interesting and the logic is sound.

---

### Meta-Review · Area_Chair_Uysm · 2024-06-16

**Recommendation:** 3

**Metareview:**

The paper presents an innovative approach that leverages LLMs to predict where SGD converges, building on prior research that links SGD to Markovian processes. The authors demonstrate that LLMs can estimate the multi-variate transition rules of SGD, which is a conceptually intriguing and novel idea. The method is both novel and promising, with clear implications for advancing the field. While the paper is already strong, it could be further enhanced by addressing the validity of the uni-variate approximation for high-dimensional SGD and providing more theoretical justification for the interpolation method used for the transition matrix. Additionally, expanding the experimental section with more Markov chains and discussing the practical applications of this approach would add even more value. Overall, this paper makes a significant and valuable contribution to the field and is highly recommended for acceptance.

---

> ### Author Response · Authors · 2024-06-28
>
> First of all, we would like to thank the reviewers and meta-reviewer for their informative and insightful feedback.
>
> Since the initial submission, we have explored several of the research directions mentioned by the reviewers and obtained novel results for them. These include both experimental results -- on markov chains, problems with more parameters, qualitative and quantitative studies -- and theoretical results justifying the interpolation method and the spectral gap in the scaling law.
>
> We will include them in the camera-ready version and are looking forward to sharing these new insights with the attendees of the workshop!

---

### Decision · Program_Chairs · 2024-06-17

Accept (Poster)